# An Analytical Model for the Normal Contact Stiffness of Mechanical Joint Surfaces Based on Parabolic Cylindrical Asperities

**DOI:** 10.3390/ma16051883

**Published:** 2023-02-24

**Authors:** Yuzhu Bai, Qi An, Shuangfu Suo, Weikun Wang, Xiaohong Jia

**Affiliations:** 1State Key Laboratory of Tribology in Advanced Equipment, Tsinghua University, Beijing 100084, China; 2School of Mechanical Electronic & Information Engineering, China University of Mining & Technology-Beijing, Beijing 100083, China

**Keywords:** plane-cutting surface, asperity, hypothetical surface, normal contact stiffness model, mechanical joint surface

## Abstract

The analytical results of normal contact stiffness for mechanical joint surfaces are quite different from the experimental data. So, this paper proposes an analytical model based on parabolic cylindrical asperity that considers the micro-topography of machined surfaces and how they were made. First, the topography of a machined surface was considered. Then, the parabolic cylindrical asperity and Gaussian distribution were used to create a hypothetical surface that better matches the real topography. Second, based on the hypothetical surface, the relationship between indentation depth and contact force in the elastic, elastoplastic, and plastic deformation intervals of the asperity was recalculated, and the theoretical analytical model of normal contact stiffness was obtained. Finally, an experimental test platform was then constructed, and the numerical simulation results were compared with the experimental results. At the same time, the numerical simulation results of the proposed model, the J. A. Greenwood and J. B. P. Williamson (GW) model, the W. R. Chang, I. Etsion, and D. B. Bogy (CEB) model, and the L. Kogut and I. Etsion (KE) model were compared with the experimental results. The results show that when roughness is *Sa* 1.6 μm, the maximum relative errors are 2.56%, 157.9%, 134%, and 90.3%, respectively. When roughness is *Sa* 3.2 μm, the maximum relative errors are 2.92%, 152.4%, 108.4%, and 75.1%, respectively. When roughness is *Sa* 4.5 μm, the maximum relative errors are 2.89%, 158.07%, 68.4%, and 46.13%, respectively. When roughness is *Sa* 5.8 μm, the maximum relative errors are 2.89%, 201.57%, 110.26%, and 73.18%, respectively. The comparison results demonstrate that the suggested model is accurate. This new method for examining the contact characteristics of mechanical joint surfaces uses the proposed model in conjunction with a micro-topography examination of an actual machined surface.

## 1. Introduction

The contact stiffness of mechanical joint surfaces plays an important role in the stability of mechanical systems [1]. Due to the complexity of the machining process of machine parts, contact-joint surfaces of mechanical parts are not completely smooth [2]. Contact of joint surfaces is the contact between asperities on rough surfaces, which will result in the actual contact area of joint surfaces being very small [3]. Therefore, contact pressure and contact stress of the asperities on joint surfaces are relatively high. From a microscopic point of view, contact between two actual surfaces is the contact of the asperities on rough surfaces [4]. For the above reasons, the actual contact pressure is much greater than the nominal contact pressure. Surface roughness has a great influence on the fatigue, wear, and other related properties of mechanical parts [5]. In the process of engineering applications, the micro-topography of mechanical parts determines the performance of the whole machine, especially in the contact of key parts; the contact stiffness of a joint surface usually presents nonlinear characteristics. Therefore, it is of engineering significance to explore the contact stiffness parameters of joint surfaces.

Research on the stiffness of the whole machine system must start from the normal contact stiffness of the mechanical joint surfaces. The analysis methods of contact stiffness mainly include the statistical analysis method and the fractal analysis method. The study of contact stiffness based on the statistical analysis method was first initiated by the GW model proposed by Greenwood and Williamson [6] in 1966. They combined statistical theory and Hertz’s contact theory to study the contact problem of rough surfaces. They assumed that all rough-surface asperities are isotropic elastic hemispheres with the same radius of curvature in contact with a smooth rigid plane. These spherical asperities are independent of each other and highly obey Gaussian distribution. Chang [7] proposed an elastoplastic convex contact model for rough surfaces, namely the CEB model, based on the control of volume conservation of asperities in the process of plastic deformation. The numerical results of this model are compared with existing pure elastic and plastic models. Some of the results have an obvious deviation from previous analysis as they do not consider the conservation of rough volume. Kogut and Etsion [8] established a frictionless elastic–plastic finite element model (KE model) of a plastic sphere under compression by a rigid plane. The model analyzes the evolution of the elastoplastic contact interface when interference increases and reveals three different stages from fully elastic to elastoplastic and then to a fully plastic contact interface. The dimensionless expressions of contact force, contact area, and average contact pressure are provided, covering a wide range of interference values from initial yield to a completely plastic spherical contact region. Compared with previous elastoplastic models based on arbitrary assumptions, the results show great differences. Based on the GW model, related scholars have carried out much research on spherical asperity, elliptical asperity, two-dimensional sinusoidal curve asperity, three-dimensional sinusoidal curve asperity, and other different shapes. Majumdar and Bhushan [9] used the fractal function (WM function) to characterize surface topography and proposed a contact model based on fractal geometry theory for the first time, namely the MB model. Yuqin Wen [10] used the watershed algorithm to segment and determine the concave–convex surface of rough surfaces and carried out ellipsoid fitting of a flange according to the principle of minimum mean-square error. Based on the elastic–plastic contact model of a single ellipsoidal asperity, a stable and efficient 3D contact analysis method for rough surfaces was proposed. Fengwei Sun [11] adopted the discrete dislocation plastic simulation method to model a rough surface as a group of equidistant convex surface arrays with sinusoidal profiles. The influence of the interaction between adjacent flanges on the contact pressure of rigid platens on rough surfaces was studied using the discrete dislocation plastic simulation method. Rostami and Jackson [12] used the finite element method to characterize the mean surface separation between a sinusoidal surface and a planar rigid surface as the mean contact pressure or force function. The finite element results also agree well with a limit ball base solution at low-contact pressure and another asymptotic solution at high-contact pressure. This fitting method can approximately predict the mean surface separation of elastic and elastoplastic sinusoidal contacts in a wide range of material properties and can predict the contact stiffness of joint surfaces. Wang [13] analyzed the loading and unloading process of a friction cylindrical surface and established a contact fractal model of spherical asperity, and deduced a nonlinear relationship between actual contact area and contact force at different deformation stages in a loading–unloading cycle.

Wang [14] established a fractal prediction model for unloading between rough contact surfaces by considering the friction coefficient between surfaces, the three-dimensional fractal characteristics of rough surfaces, and the elastic–plastic deformation mechanism of convex surfaces. The friction coefficient and the fractal dimension of the contact surface affect the unloading model. The unloading process between rough surfaces depends on the final loading state of the rough surface. The energy dissipation in the contact cycle causes the loading and unloading curves to form a hysteretic loop. Wang [15] adopted the Weierstrass–Mandelbrot function in fractal theory to represent a contact-parameter analysis method of rough contact surfaces and established a fretting damage model based on fractal theory. It is proved that the fitting accuracy of the fractal parameters based on the neural network algorithm is better than that of the traditional method. Yuan [16] established an elastic–plastic contact model for the loading and unloading of three-dimensional fractal rough surfaces. The total contact area and total contact load during loading and unloading were obtained. When the rough surface is under elastic deformation, the load–area relationship is the same during loading and unloading. When the rough surface is under inelastic deformation, the real contact area during unloading is larger than that during loading. Yu [17] believed that when the topography parameters of two rough surfaces are similar, shoulder–shoulder contact should be adopted instead of top–top contact. Based on shoulder contact and fractal characteristics, a convex geometric model and rough-surface contact mechanics model were established. Zhou [18] used the maximum valley–peak ratio (VPR) to establish a three-dimensional concave–convex model for rough surfaces. The convex function of a paraboloid was derived by the least-squares method. A convex projection plane was determined by the VPR method, and the rough convex surface was reconstructed. Zhang [19] used the finite element method to conduct a deterministic contact analysis of two three-dimensional rough surfaces based on artificially generated rough surfaces. The calculation results show that when the surface spacing decreases, the results obtained by the classical-model-based method are quite different from those obtained in the paper, which is more obvious when considering roughness distribution and texture. Wen [20] considered the interaction of concave–convex surfaces and elastic–plastic deformation characteristics, and further proposed an analytical calculation method for rough-surface contact considering the interaction of concave–convex surfaces. Based on the watershed algorithm, a rough surface was segmented, and an ellipsoid model was established. Yu [21] converted an elastic contact problem between gear-meshing surfaces into a contact problem between elastic surfaces with arbitrary curvature radius, and proposed a contact-area distribution function for the convex surface of a curved point-contact ellipse. On this basis, a fractal contact mechanics model of elliptic-bevel-gear rough surfaces was established. The influence of tooth-surface topography on contact load and contact stiffness under different fractal parameters was studied. The results show that contact stiffness and actual contact load increase with an increase in contact coefficient and fractal dimension. Normal contact stiffness and actual contact load decrease with an increase in eccentricity and fractal roughness. Zhang and Ren [22] proposed a method for calculating the actual contact area and load under progressive indentation depths of general rough surfaces. A three-dimensional finite element model of elliptic asperity for analyzing mechanical contact problems was established, and the results of an electromechanical coupling simulation were compared with the new calculation method. The high aspect ratio of this method is beneficial to accurately calculate contact area under micro-contact conditions. Li [23] used the Fourier series to separate surface roughness and corrugability, established a rough-surface contact model, and obtained the displacement caused by the interaction of rough surfaces. In this paper a joint surface was regarded as the contact between a rough surface and a smooth corrugated surface, and a new model considering the interaction of rough surfaces and surface corrugations was obtained. Wang and Schipper [24] improved the accuracy of the semi-analytical method (SAM) to calculate actual contact area using an analytically generated sine-wave surface to simulate a real rough surface on a rigid plane. Wang [25] extracted shape-measurement points through harmonic interpolation to construct a three-dimensional deterministic model of a rough surface.

Research on mechanical joint surfaces should be based on actual topography; otherwise, the normal contact stiffness will be separated from the actual working condition. There is a certain gap between the hypothetical surface and the actual surface, or, to some extent, it can only be close to a certain type of rough surface. There is a great difference between actual topography under different machining methods and spherical or elliptic asperity assumptions, such as turning, milling, and grinding. As a result, there is still no universal model for a wide range of applications. In this paper, based on surface-topography data of various machining methods, a hypothetical surface that can more accurately represent the topography characteristics of an actual cutting surface was established. Therefore, a hypothetical surface of asperities with a cross-sectional profile of parabola Y = CX^2^ + GX + H is established. Based on this hypothetical surface, the relationship between the contact parameters and contact force in the contact area is deduced by the theory of statistics and contact mechanics. Then, a type of contact model that is more suitable for cutting rough surfaces is established. The innovation point of this paper is to establish a parabolic cylindrical asperity model with a cross-sectional contour by analyzing the combined surfaces obtained by various machining methods. Compared with other models, this new model is closer to real contact conditions, and the comparison between experimental results and simulation results shows that the accuracy of this model is very high. The new model is obtained by summarizing all types of binding surface and has a wider application.

## 2. Establishment of the Hypothetical Surface

From a microscopic perspective, contact of mechanical joint surfaces is the contact of the asperities on rough surfaces. Therefore, a contact model should be studied from the perspective of asperity contact. In this section, data for rough surfaces will be collected using a white-light interferometer, and the contour parameters of a single asperity will be fitted by combining the topography data measured. Then, according to the above fitting data, a new hypothetical surface will be established.

To obtain actual topography data of rough surfaces, a 3D white-light interferometer (ZYGONexView, ZYGO Connecticut, Middlefield, CT, USA) was used to measure the surface topography of workpieces. The device adopts the principle of non-contact white-light interference, closed-loop feedback piezoelectric ceramic, and a high linear capacitance sensor, which can ensure the full range of a 0–150 μm 0.1 nm high-precision test, to ensure the real validity of the test results. Figure 1 shows the topography of surface roughness obtained by boring, turning, smooth grinding, and grinding methods, which are *Sa* 3.031 μm, *Sa* 2.418 μm, *Sa* 0.076 μm, and *Sa* 0.037 μm, respectively.

As shown in Figure 1, surface grooves of workpieces obtained by different processing methods have a clear texture: an obvious corrugated undulation shape and a uniform distribution of concave and convex surfaces. In general, the surface topography obtained by the above processing method is more similar to a well-aligned cylindrical body with a similar cross-sectional curvature radius, rather than a spherical or parabolic rotator asperity. The 3D and 2D topographies are analyzed in combination with the micro-topographies of the boring surface. Figure 2 shows the 3D and 2D reconstructed topographies in the boring machining mode based on measured data.

Figure 2a shows the 3D reconstructed topography of boring machining based on measured data. It can be seen that the 3D reconstruction is consistent with the actual topography. Moreover, asperities in the Y direction show a consistent feature. Characteristics of periodic distribution are seen in the X direction, but asperity height is not consistent. Therefore, the study of three-dimensional surface topography can be simplified to that of a two-dimensional surface. Figure 2b shows the two-dimensional topography analysis diagram of an XOZ section. The peak–valley labeling method [26] was used for statistical analysis of asperity height, and the results showed that the height distribution of the asperities presented a Gaussian distribution.

The following is a discussion of the shape of a single asperity. A data-fitting method is adopted to process randomly selected single-asperity-measured data points. Figure 3 shows the fitting results of a single asperity shape-profile data point. The fitted surface retains the real texture structure and height features of the original surface and accurately reflects the real state of the original surface. To reveal the overall characteristics of the actual profile of the asperity, the parabolic function Y = CX^2^ +GX + H was used to fit the data points of a single asperity profile, and the fitting root-means-square error was 0.0002667.

It can be seen from the fitting results that the parabolic curve can be completely consistent with the measured profile data of the asperity. Therefore, based on the more accurate fitting of the rough-surface topography, the parabolic cylinder is used to represent the shape of the single asperity. The height distribution of asperities is represented by a Gaussian distribution.

In summary, based on the analysis of the micro-topography of machined surfaces, a hypothetical surface based on parabolic cylindrical asperity was proposed for the mechanical joint surface in this paper. A single surface asperity is represented by a parabolic column shape, and the height distribution of the asperities is Gaussian. The established hypothetical surface is shown in Figure 4.

## 3. The Analytical Model of Normal Contact Stiffness

### 3.1. Contact Model Hypothesis

In Section 2, a hypothetical surface based on parabolic cylindrical asperity is proposed by analyzing the measured surface topography. In this section, a new theoretical analytical model of normal contact stiffness is established based on the hypothetical surface. Before establishing the analytical model, the following assumptions were made in this paper for the theoretical contact model of asperities on rough surfaces: (1) plane contact takes two cross-sectional profile curves as parabolic Y = CX^2^ + GX + H cylinder contact, and the cross angle of the main axis of the cylinder is 45° on average; (2) the curvature radius of all asperity surfaces on the contact surface is the same, and the height of asperity surfaces follows a Gaussian distribution; (3) the deformation of asperities is independent of each other, and the interaction between asperities is ignored in the contact process; (4) only the asperities deform, while the matrix does not participate in deformation; (5) the asperities are uniformly distributed on the matrix. Figure 5 shows a schematic diagram of the model hypothesis surface.

Based on the contact model established above, this section first analyzes the stress and deformation of two separate asperities in contact with each other. The theoretical formula between the contact force and deformation is solved, and the theoretical normal contact stiffness is obtained. Combined with statistical theory, contact force and deformation formulae for the whole mechanical joint surface were derived, and then a theoretical analytical model of the normal contact stiffness of the whole joint surface was obtained.

### 3.2. Contact Stiffness Analysis of a Single Asperity

The contact process of an asperity will undergo three stages: elastic deformation, elastoplastic deformation, and complete plastic deformation. The following will be a separate analysis of the different deformation stages.

Elastic deformation

According to the Hertz contact theory, the contact surface of two parabolic cylinders Y = CX^2^ +GX + H is an elliptic region, and the short diameter a and long diameter b of the ellipse are calculated by Equations (1) and (2) [27].
(1)$$a=\alpha {\left[\frac{3}{4}\frac{F}{A+B}\left(\frac{1-\nu_{1}^{2}}{E_{1}}+\frac{1-\nu_{2}^{2}}{E_{2}}\right)\right]}^{\frac{1}{3}}$$
(2)$$b=\beta {\left[\frac{3}{4}\frac{F}{A+B}\left(\frac{1-\nu_{1}^{2}}{E_{1}}+\frac{1-\nu_{2}^{2}}{E_{2}}\right)\right]}^{\frac{1}{3}}$$
where *F* is the concentrated force; *ν*_1_ and *ν*_2_ are the Poisson’s ratios of materials, respectively; and *E*_1_ and *E*_2_ are the elastic moduli of materials, respectively. The influence coefficients A and B of the principal curvature can be obtained by Equations (3) and (4) [27].
(3)$$B+A=\frac{1}{2}\left(\frac{1}{R_{1}}+\frac{1}{R_{1}\prime }+\frac{1}{R_{2}}+\frac{1}{R_{2}\prime }\right)$$
(4)$$B-A=\frac{1}{2}{\left[{\left(\frac{1}{R_{1}}-\frac{1}{R_{1}\prime }\right)}^{2}+{\left(\frac{1}{R_{2}}-\frac{1}{R_{2}\prime }\right)}^{2}+2\left(\frac{1}{R_{1}}-\frac{1}{R_{1}\prime }\right)\left(\frac{1}{R_{2}}-\frac{1}{R_{2}\prime }\right)\cos2\varphi \right]}^{\frac{1}{2}}$$
where *R*_1_ and *R*_1_ʹ are object 1 at the contact points of the two main curvature radii, respectively; *R*_2_ and *R*_2_ʹ are object 2 at the contact points of the two main curvature radii, respectively; *φ* is the angle between the planes where the two main curvature radii *R*_1_ and *R*_2_ are located. It is assumed that the contact of two rough surfaces is the contact of two parabolic cylindrical asperities with the same radius of curvature. The mean value of the intersection angle φ of the axes of two asperities is *φ* = 45°. The radius of curvature at a point on the curve is reciprocal to the curvature of the curve at that point. The radial radii *R*_1_ and *R*_2_ of the main curvatures of the asperity are equal to the curvature radius *R* of the parabola, and their curvatures 1/*R*_1_ and 1/*R*_2_ are also equal: that is, 1/*R*_1_ = 1/*R*_2_ = 1/*R*. The axial main radii of curvatures *R*_1_ʹ and *R*_2_ʹ to infinity obtain curvatures 1/*R*_1_ʹ and 1/*R*_2_ʹ that are equal to zero; that is, 1/*R*_1_ʹ = 1/*R*_2_ʹ = 0. By combining Equations (3) and (4), we can obtain $A=1/2R-1/2\sqrt{2}R$ and *B*$=1/2R+1/2\sqrt{2}R$. We calculate the curvature of the parabolic curve Y = CX^2^ +GX + H according to the curvature formula, as shown in Equation (5).
(5)$$D=\frac{\left|Y\prime \prime \right|}{{\left(1+Y\prime^{2}\right)}^{\frac{3}{2}}}=\frac{\left|2C\right|}{{\left(4C^{2}X^{2}+4CGX+G^{2}+1\right)}^{\frac{3}{2}}}$$

Meanwhile, the radius of the main curvature can be determined according to Equation (5).
(6)$$R=\frac{1}{D}=\frac{{\left(4C^{2}X^{2}+4CGX+G^{2}+1\right)}^{\frac{3}{2}}}{\left|2C\right|}$$

It is assumed that only elastic deformation occurs when two asperities are first in contact. According to the Hertz contact theory, the relationship between proximity distance ω and force F of the center of two asperities can be obtained.
(7)$$\omega =\lambda {\left[\frac{9}{128}F^{2}\left(A+B\right){\left(\frac{1-\nu_{1}^{2}}{E_{1}}+\frac{1-\nu_{2}^{2}}{E_{2}}\right)}^{2}\right]}^{\frac{1}{3}}$$

Equation (7) [27] is the elastic-contact Hertz formula of two parabolic column surfaces with the same radius of principal curvature. In the equation, the coefficients α, β, and λ can be obtained from setting *θ_ο_* = 45° and then consulting the elastic force manually to obtain *α* = 1.926, *β* = 0.604, and *λ* = 1.709.
(8)$$\omega =\lambda {\left[\frac{9}{128}\frac{F^{2}}{R}{\left(\frac{1-\nu_{1}^{2}}{E_{1}}+\frac{1-\nu_{2}^{2}}{E_{2}}\right)}^{2}\right]}^{\frac{1}{3}}$$

The relationship between connection displacement ω and force *F* at the stage of pure elastic deformation is derived from Equation (8):(9)$$F\left(\omega \right)=\frac{8\sqrt{2R\omega^{3}}}{3\lambda^{\frac{3}{2}}\left(\frac{1-\nu_{1}^{2}}{E_{1}}+\frac{1-\nu_{2}^{2}}{E_{2}}\right)}$$

Contact stiffness is the derivative of the contact force concerning displacement. Thus, contact stiffness in the pure-elastic-contact stage can be expressed as:(10)$$K=\frac{dF\left(\omega \right)}{d\omega }=\frac{4\sqrt{2R\omega }}{\lambda^{\frac{3}{2}}\left(\frac{1-\nu_{1}^{2}}{E_{1}}+\frac{1-\nu_{2}^{2}}{E_{2}}\right)}$$

2.Elastoplastic deformation

According to the contact mechanics theory of Tabor [28], when the average contact pressure of two rough surfaces reaches *P_a_* = 0.4 *H*, an asperity on the joint surface begins to yield and deform, thus:(11)$$P_{e}=\frac{F_{1}}{\pi ab}=0.4H$$
where *H* is material hardness. By substituting Equations (1) and (2) into Equation (11), we obtain:(12)$$\frac{F_{1}}{\pi \alpha {\left[\frac{3}{4}\frac{F_{1}}{A+B}\left(\frac{1-\nu_{1}^{2}}{E_{1}}+\frac{1-\nu_{2}^{2}}{E_{2}}\right)\right]}^{\frac{1}{3}}\beta {\left[\frac{3}{4}\frac{F_{1}}{A+B}\left(\frac{1-\nu_{1}^{2}}{E_{1}}+\frac{1-\nu_{2}^{2}}{E_{2}}\right)\right]}^{\frac{1}{3}}}=0.4H$$

The force at the onset of plastic deformation of the asperity is obtained after the end of displacement:(13)$$F_{1}=0.036{\left(H\pi \alpha \beta \right)}^{3}R^{2}{\left(\frac{1-\nu_{1}^{2}}{E_{1}}+\frac{1-\nu_{2}^{2}}{E_{2}}\right)}^{2}$$

Then displacement *ω*_1_ at the onset of plastic deformation of the asperity is derived:(14)$$\omega_{1}=1.709{\left[\frac{9}{128R}F_{1}{}^{2}{\left(\frac{1-\nu_{1}^{2}}{E_{1}}+\frac{1-\nu_{2}^{2}}{E_{2}}\right)}^{2}\right]}^{\frac{1}{3}}$$

According to the literature [29], the contact area at the elastoplastic deformation stage is:(15)$$S_{ep}=\pi \alpha \beta {\left[\frac{3}{4}\frac{F_{ep}}{A+B}\left(\frac{1-\nu_{1}^{2}}{E_{1}}+\frac{1-\nu_{2}^{2}}{E_{2}}\right)\right]}^{\frac{2}{3}}\left[1-2{\left(\frac{\omega -\omega_{1}}{\omega_{2}-\omega_{1}}\right)}^{3}+3{\left(\frac{\omega -\omega_{1}}{\omega_{2}-\omega_{1}}\right)}^{2}\right]$$

Average contact pressure during elastoplastic deformation is:(16)$$P_{ep}=H-0.6H\frac{\ln\omega_{2}-\ln\omega }{\ln\omega_{2}-\ln\omega_{1}}$$

Thus, the contact force at the elastoplastic deformation stage is obtained:(17)$$F_{ep}=P_{ep}S_{ep}=\left(H-0.6H\frac{\ln\omega_{2}-\ln\omega }{\ln\omega_{2}-\ln\omega_{1}}\right)\times \pi \alpha \beta {\left[\frac{3}{4}\frac{F_{ep}}{A+B}\left(\frac{1-\nu_{1}^{2}}{E_{1}}+\frac{1-\nu_{2}^{2}}{E_{2}}\right)\right]}^{\frac{2}{3}}\left[1-2{\left(\frac{\omega -\omega_{1}}{\omega_{2}-\omega_{1}}\right)}^{3}+3{\left(\frac{\omega -\omega_{1}}{\omega_{2}-\omega_{1}}\right)}^{2}\right]$$

So,
(18)$$F_{ep}=\frac{9}{16}{\left(\pi \alpha \beta \right)}^{3}{\left[R\left(\frac{1-\nu_{1}^{2}}{E_{1}}+\frac{1-\nu_{2}^{2}}{E_{2}}\right)\right]}^{2}{\left(H-0.6H\frac{\ln\omega_{2}-\ln\omega }{\ln\omega_{2}-\ln\omega_{1}}\right)}^{3}{\left[1-2{\left(\frac{\omega -\omega_{1}}{\omega_{2}-\omega_{1}}\right)}^{3}+3{\left(\frac{\omega -\omega_{1}}{\omega_{2}-\omega_{1}}\right)}^{2}\right]}^{3}$$

Contact stiffness is the derivative of contact force with respect to displacement. Thus, the expression of contact stiffness in the elastic–plastic contact stage can be obtained as follows:(19)$$\begin{array}{ll} K_{ep}= & \frac{27}{16}{\left(\pi \alpha \beta \right)}^{3}{\left[R\left(\frac{1-\nu_{1}^{2}}{E_{1}}+\frac{1-\nu_{2}^{2}}{E_{2}}\right)\right]}^{2}\left(\frac{0.6H}{\omega \left(\ln\omega_{2}-\ln\omega_{1}\right)}\right){\left(H-0.6H\frac{\ln\omega_{2}-\ln\omega }{\ln\omega_{2}-\ln\omega_{1}}\right)}^{2}[1-2{\left(\frac{\omega -\omega_{1}}{\omega_{2}-\omega_{1}}\right)}^{3}\\ & {+3{\left(\frac{\omega -\omega_{1}}{\omega_{2}-\omega_{1}}\right)}^{2}]}^{3}+\frac{27}{16}{\left(\pi \alpha \beta \right)}^{3}{\left[R\left(\frac{1-\nu_{1}^{2}}{E_{1}}+\frac{1-\nu_{2}^{2}}{E_{2}}\right)\right]}^{2}{\left(H-0.6H\frac{\ln\omega_{2}-\ln\omega }{\ln\omega_{2}-\ln\omega_{1}}\right)}^{3}[1-2{\left(\frac{\omega -\omega_{1}}{\omega_{2}-\omega_{1}}\right)}^{3}\\ & {+3{\left(\frac{\omega -\omega_{1}}{\omega_{2}-\omega_{1}}\right)}^{2}]}^{2}\left[\frac{6\left(\omega -\omega_{1}\right)}{{\left(\omega_{2}-\omega_{1}\right)}^{2}}-\frac{6{\left(\omega -\omega_{1}\right)}^{2}}{{\left(\omega_{2}-\omega_{1}\right)}^{3}}\right] \end{array}$$

According to the literature [8], when deformation at the microscopic contact point reaches 110 times the initial yield deformation, the asperity enters a stage of complete plastic deformation; that is, the critical deformation from elastoplastic to complete plastic deformation is ω_2_ = 110 ω_1_.

3.Complete plastic deformation

According to the viewpoints of Abbott and Firestone [30], the relationship between contact area and displacement at the stage of complete plastic deformation can be expressed as:(20)$$S_{p}=2\pi R\omega =2\pi ab$$

According to the contact mechanics theory of Tabor [28], when the average contact pressure reaches *P_a_* = *H*, the asperity begins to yield completely.
(21)$$F_{p}=S_{p}P_{a}=2\pi R\omega H=2\pi \alpha \beta {\left[\frac{3}{4}\frac{F_{p}}{A+B}\left(\frac{1-\nu_{1}^{2}}{E_{1}}+\frac{1-\nu_{2}^{2}}{E_{2}}\right)\right]}^{\frac{2}{3}}H$$
(22)$$F_{p}=\frac{9}{2}{\left(\pi \alpha \beta \right)}^{3}{\left[R\left(\frac{1-\nu_{1}^{2}}{E_{1}}+\frac{1-\nu_{2}^{2}}{E_{2}}\right)\right]}^{2}H^{3}$$

Contact stiffness is the derivative of the contact force concerning displacement. Thus, the expression of contact stiffness at the stage of complete plastic deformation is:(23)$$K_{p}=\frac{dF_{p}\left(\omega \right)}{d\omega }=2\pi RH$$

### 3.3. Contact Stiffness of Mechanical Joint Surfaces

Based on the analysis in Section 3.2, the contact stiffness of a single asperity in different deformation intervals during contact is obtained. In this section, the contact stiffness of the whole joint surface will be solved by combining statistical theory. According to the analysis in Section 2, the heights of the parabolic cylindrical asperities on a rough surface follow a Gaussian distribution. The contact stiffness of an asperity in the whole contact region can be obtained by integrating it with the distribution function.

When upper and lower rough surfaces contact, assume that the distance between the average height line of asperities on the two rough surfaces is *l*, and take the average height line of asperities on the lower rough surface as the origin of the *z*-axis. All asperities with height z>l/2=d participate in the contact deformation. The pressing depth of an asperity with height *z* is *h* = *z* − *d*. Therefore, the probability of contact at any given height of a rough surface can be obtained from the equation below.
(24)$$\mathrm{Pr}ob=\left(z>d\right)=\int_{d}^{\infty }\frac{1}{\sqrt{2\pi }\sigma }e^{-\frac{{\left(z\right)}^{2}}{2\sigma^{2}}}dz$$

If the total number of asperities on the entire contact surface is *N*, then the number of asperities participating in the contact is:(25)$$n=N\int_{d}^{\infty }\frac{1}{\sqrt{2\pi }\sigma }e^{-\frac{{\left(z\right)}^{2}}{2\sigma^{2}}}dz$$

Meanwhile, since *ω* = 2 *h* = 2(*z* − *d*), the total force *F* can be derived.
(26)$$F=N\int_{d}^{\infty }F\left(2z-2d\right)\frac{1}{\sqrt{2\pi }\sigma }e^{-\frac{{\left(z-d\right)}^{2}}{2\sigma^{2}}}dz$$

Meanwhile, the total contact stiffness can be obtained by the following equation.
(27)$$K=N\int_{d}^{\infty }K\left(2z-2d\right)\frac{1}{\sqrt{2\pi }\sigma }e^{-\frac{{\left(z-d\right)}^{2}}{2\sigma^{2}}}dz$$

The following derivation is made based on the contact stiffness of the joint surface at different deformation stages.

Elastic deformation

The expression of the force at the elastic deformation stage is:(28)$$F_{e}\left(z\right)=N\int_{d}^{d+\frac{\omega_{1}}{2}}\frac{32\sqrt{R{\left(z-d\right)}^{3}}}{3\lambda^{\frac{3}{2}}\left(\frac{1-\nu_{1}^{2}}{E_{1}}+\frac{1-\nu_{2}^{2}}{E_{2}}\right)}\frac{1}{\sqrt{2\pi }\sigma }e^{-\frac{{\left(z-d\right)}^{2}}{2\sigma^{2}}}dz$$

The expression of contact stiffness at the elastic deformation stage is:(29)$$K_{e}\left(z\right)=\frac{dF\left(z\right)}{dz}=N\int_{d}^{d+\frac{\omega_{1}}{2}}\frac{8\sqrt{R\left(z-d\right)}}{\lambda^{\frac{3}{2}}\left(\frac{1-V_{1}^{2}}{E_{1}}+\frac{1-V_{2}^{2}}{E_{2}}\right)}\frac{1}{\sqrt{2\pi }\sigma }e^{-\frac{{\left(z-d\right)}^{2}}{2\sigma^{2}}}dz$$

2.Elastoplastic deformation

The expression of the contact force at the elastoplastic deformation stage is:(30)$$\begin{array}{ll} F_{ep}\left(z\right)= & N\int_{d+\frac{\omega_{1}}{2}}^{d+\frac{\omega_{2}}{2}}\frac{9}{16}{\left(\pi \alpha \beta \right)}^{3}{\left[R\left(\frac{1-\nu_{1}^{2}}{E_{1}}+\frac{1-\nu_{2}^{2}}{E_{2}}\right)\right]}^{2}{\left(H-0.6H\frac{\ln\omega_{2}-\ln2\left(z-d\right)}{\ln\omega_{2}-\ln\omega_{1}}\right)}^{3}\cdot \\ & [1-2{\left(\frac{2z-2d-\omega_{1}}{\omega_{2}-\omega_{1}}\right)}^{3}+{3{\left(\frac{2z-2d-\omega_{1}}{\omega_{2}-\omega_{1}}\right)}^{2}]}^{3}\frac{1}{\sqrt{2\pi }\sigma }e^{-\frac{{\left(z-d\right)}^{2}}{2\sigma^{2}}}dz \end{array}$$

The expression of contact stiffness at the elastoplastic deformation stage is
(31)$$\begin{array}{ll} K_{ep}= & \frac{27}{16}N\int_{d+\frac{\omega_{1}}{2}}^{d+\frac{\omega_{2}}{2}}{\left(\pi \alpha \beta \right)}^{3}{\left[R\left(\frac{1-\nu_{1}^{2}}{E_{1}}+\frac{1-\nu_{2}^{2}}{E_{2}}\right)\right]}^{2}\left[\frac{0.6H}{2\left(z-d\right)\left(\ln\omega_{2}-\ln\omega_{1}\right)}\right]\cdot \\ & {\left[H-0.6H\frac{\ln\omega_{2}-\ln2\left(z-d\right)}{\ln\omega_{2}-\ln\omega_{1}}\right]}^{2}[1-2{\left(\frac{2z-2d-\omega_{1}}{\omega_{2}-\omega_{1}}\right)}^{3}{+3{\left(\frac{2z-2d-\omega_{1}}{\omega_{2}-\omega_{1}}\right)}^{2}]}^{3}\\ & +{\left(\pi \alpha \beta \right)}^{3}{\left[R\left(\frac{1-\nu_{1}^{2}}{E_{1}}+\frac{1-\nu_{2}^{2}}{E_{2}}\right)\right]}^{2}{\left[H-0.6H\frac{\ln\omega_{2}-\ln2\left(z-d\right)}{\ln\omega_{2}-\ln\omega_{1}}\right]}^{3}[1-2{\left(\frac{2z-2d-\omega_{1}}{\omega_{2}-\omega_{1}}\right)}^{3}\\ & {+3{\left(\frac{2z-2d-\omega_{1}}{\omega_{2}-\omega_{1}}\right)}^{2}]}^{2}\left[\frac{6\left(2z-2d-\omega_{1}\right)}{{\left(\omega_{2}-\omega_{1}\right)}^{2}}-\frac{6{\left(2z-2d-\omega_{1}\right)}^{2}}{{\left(\omega_{2}-\omega_{1}\right)}^{3}}\right]\frac{1}{\sqrt{2\pi }\sigma }e^{-\frac{{\left(z-d\right)}^{2}}{2\sigma^{2}}}dz \end{array}$$

3.Complete plastic deformation

The expression of the force at the stage of complete plastic deformation is:(32)$$F_{p}\left(z\right)=4\pi NRH\int_{d+\frac{\omega_{2}}{2}}^{\infty }(z-d)\frac{1}{\sqrt{2\pi }\sigma }e^{-\frac{{\left(z-d\right)}^{2}}{2\sigma^{2}}}dz$$

The contact stiffness at the stage of complete plastic deformation can be expressed as:(33)$$K_{p}\left(z\right)=2\pi NRH\int_{d+\frac{\omega_{2}}{2}}^{\infty }\frac{1}{\sqrt{2\pi }\sigma }e^{-\frac{{\left(z-d\right)}^{2}}{2\sigma^{2}}}dz$$
where $\Phi \left(z\right)=\frac{1}{\sqrt{2\pi }\sigma }e^{-\frac{{\left(z-d\right)}^{2}}{2\sigma^{2}}}$ is a Gaussian distribution function.

## 4. Experimental Assessment

### 4.1. The Experimental Specimen

The experimental specimen is made of 45 steel (Carbon (C) content is 0.42~0.50%; Si content is 0.17–0.37%; Mn content is 0.50~0.80%; Cr content is ≤0.25%; Ni content is ≤ 0.30%; Cu content is ≤ 0.25%), and the parameters of 45 steel (Young’s modulus *E*_1_ = *E*_2_ = 209 GPa, Poisson’s ratio ν_1_ = ν_2_ = 0.269, hardness *H* = 1970 MPa) are obtained by conventional mechanical experiments. As shown in Figure 6 below, the joint surface of the normal contact stiffness-test specimen was obtained by an end-turning process, and the height of the convex platform was 1 mm. The convex platform roughness of the specimen joint surface can be divided into four values: *Sa* 1.6μm, *Sa* 3.2 μm, *Sa* 4.5 μm, *Sa* 5.8 μm; and each roughness has four diameters of 10 mm, 20 mm, 30 mm, and 40 mm. The lower specimen has a diameter of 40 mm. So that the upper and lower roughness specimens are consistent, the lathe and turning tool for each type of roughness specimen are consistent, and the machine tool’s spindle speed, turning angle, feed speed, and turning amount are all consistent. To ensure consistent roughness on the upper and lower specimens, the lathe and turning tools are consistent for each roughness specimen, and the spindle speed, turning angle, feed speed, and turning amount of the machine tool are all consistent.

### 4.2. The Experimental Device

To verify the accuracy of the model proposed in this paper, a test platform for normal contact stiffness was built. As shown in Figure 7a, the test platform is composed of four parts: a universal testing machine, a measuring shaft system, a data-acquisition system, and a control system. The universal experimental machine is a WDW-series microcomputer-controlled electronic universal-testing machine (Shimadzu kyoto, Japan) (5–20 KN), and the relative error of the test force is less than ±0.5%. The measuring shaft system is shown in Figure 7b. The measuring shaft system consists of seven parts: loading push rod, upper test specimen, eddy-current sensor, lower test specimen, fixed disk, thrust roller bearing, and base. The eddy-current sensor is a KAMAN high-precision eddy-current displacement sensor(KAMAN Bloomfield, CT, USA), model KD2306-2U, with a measuring accuracy of ± 0.1 μm, output voltage 0–10 V, and measuring range 0–1 mm. The data-acquisition system uses an Altai data collector, model PCIe5653.

Figure 7 shows the normal contact stiffness test device for mechanical joint surfaces. The upper test specimen was fixed at the lower end of the push rod, and three eddy-current sensors were installed in three holes uniformly distributed on the upper specimen. The average displacement values of the three eddy-current sensors were taken as the test results, which effectively ensured measurement accuracy. The lower test specimen was attached to the fixed disc, and there was a thrust roller bearing under the fixed disc to adjust the installation error, ensure normal contact between the upper and lower test specimens, and offset the installation deviation. First, the loading parameters of the universal testing machine (load range, loading time, number of cycles, etc.) were set. The universal testing machine with the push rod used in the vertical direction will produce a normal thrust, and this thrust is applied to the upper test specimen through the push rod. Through the upper test specimen, a uniform action is applied to the lower test specimen. At the same time, the joint surface was subjected to a uniform distribution of static normal forces, and corresponding deformation was generated. The deformation was obtained by three eddy-current displacement sensors uniformly installed on the upper test specimen. After that, the data collector transmitted the obtained data to the computer for storage, and finally realized the collection of the static force and deformation data of the joint surface. Using the voltage data obtained by the eddy-current sensor (unit, V) converted to displacement (unit, Micron), combined with the load data (unit, N) obtained by the universal testing machine, the normal contact experimental stiffness of the joint surface could be obtained. Three tests were performed for each roughness and diameter, and the results were almost identical each time. This shows that the experimental method and test results are reliable and repeatable.

## 5. Comparative Results and Discussion

The roughness obtained by cutting is closely related to the equipment, tool, and cutting conditions. The machining process is very complicated, and the surface roughness varies greatly. Therefore, roughness is an important parameter for measuring machining accuracy and is the most important material uncertainty factor of machining surfaces. The national general roughness standards are Ra1.6, R3.2, Ra6.3, etc. Therefore, roughness values of Sa 1.6 μm, Sa 3.2 μm, Sa 4.5 μm, and Sa 5.8 μm, close to the national standard, were selected in this paper. The validity of the model is verified by comparative experiments. Experimental results of normal contact stiffness of mechanical bonding surfaces with different roughness could be obtained by the test bench built in Section 4. The variation curve of normal contact stiffness with contact pressure under different roughness is shown in Figure 8a. At the same time, to verify the correctness of the proposed model, the proposed model and experimental results with different roughness were compared and analyzed, as shown in Figure 8b.

Figure 8a shows the experimental curves of normal contact stiffness of specimens with roughness of *Sa* 1.6 μm, *Sa* 3.2 μm, *Sa* 4.5 μm, and *Sa* 5.8 μm. As shown in the figure, by comparing the normal contact stiffness curves under four different roughness conditions, it can be seen that the stiffness of the joint surface with low roughness is always greater than that of the joint surface with high roughness as the pressure increases, and normal contact stiffness also increases with an increase in normal contact pressure. Therefore, contact stiffness of the joint surface with low roughness is greater than that of the joint surface with high roughness. When the normal pressure is increased up to 6.3 MPa, the normal contact stiffnesses for roughness values of *Sa* 1.6 μm, *Sa* 3.2 μm, *Sa* 4.5 μm, and *Sa* 5.8 μm are 8.8 MPa/μm, 6.3 MPa/μm, 4.5 MPa/μm, and 3.5 MPa/μm, respectively. The reasons for this are analyzed from the perspective of mechanical principles. At the onset of loading, the pressure is relatively small, the deformation of the asperities is within the elastic range, the contact deformation of the joint surface is mainly elastic, and the relationship between the normal pressure and normal contact stiffness is approximately linear. However, with an increase in applied pressure, the asperities deform elastoplastically. The contact deformation of the joint surface is mainly elastoplastic, and the relationship between normal pressure and normal stiffness shows a nonlinear trend. When the normal contact pressure is high, the asperities begin to undergo plastic deformation, the contact deformation of the joint surface is mainly plastic, and even the substrate material of the joint surface begins to undergo deformation. The asperity deformation gradually enters a stage of complete plastic deformation. The relationship between normal pressure and normal contact stiffness appears nonlinear, the normal contact stiffness gradually tends towards a fixed value, and the slope of the curve is close to horizontal. From the perspective of surface-topography characteristics, the reasons are as follows: under the same pressure, the joint surface of the specimen with low roughness has a large density of asperities, a large number of contact asperities, and the relative deformation is smaller than that of the joint surface with high roughness, resulting in a relatively high contact stiffness value. Moreover, surface corrugability, basic asperity parameters, and average specimen height are also different, resulting in a large gap in contact stiffness. The above analysis shows that roughness has a great impact on the performance of mechanical parts. Improving machining accuracy and reducing roughness can improve the normal contact stiffness of parts and improve the stability of mechanical equipment.

Figure 8b shows the comparison between the experimental curve and the model curve of the normal contact stiffness of a joint surface at different values of roughness: *Sa* 1.6 μm, *Sa* 3.2 μm, *Sa* 4.5 μm, and *Sa* 5.8 μm. As shown in the figure, by comparing the stiffness curves under four different roughness conditions, it can be seen that in the test with a specimen roughness of *Sa* 1.6 μm, when normal pressure reaches 2.98 MPa, experimental stiffness reaches 5.62 MPa/μm. The stiffness curve of this model intersects the experimental stiffness curve. When normal pressure is less than 2.98 MPa, experimental stiffness is slightly less than the numerical simulation value of this model. When normal pressure is greater than 2.98 MPa, experimental stiffness is slightly greater than the numerical simulation value of this model. When the normal pressure is 5.09 MPa, the numerical simulation value of the stiffness of this model is 7.58 MPa/μm, and the experimental stiffness is 7.78 MPa/μm. The curve has slightly deviated, and the maximum relative error is 2.56%. The roughness is *Sa* 3.2 μm. When normal pressure reaches 3.43 MPa, experimental stiffness reaches 4.37 MPa N/μm, and the stiffness curve of this model intersects the experimental stiffness curve. When normal pressure is less than 3.43 MPa, experimental stiffness is slightly less than the numerical simulation value of this model. When normal pressure is greater than 3.43 MPa, experimental stiffness is slightly greater than the numerical simulation value of this model. When normal pressure is 1.91 MPa, the numerical simulation stiffness of the model is 3.05 MPa/μm and the experimental stiffness is 2.96 MPa/μm. The curve has slightly deviated, and the maximum relative error is 2.92%. The roughness is *Sa* 4.5 μm. When normal pressure reaches 3.82 MPa, experimental stiffness reaches 3.53 MPa/μm, and the stiffness curve of this model intersects the experimental stiffness curve. When normal pressure is less than 3.82 MPa, experimental stiffness is slightly greater than the numerical simulation value of this model, and when normal pressure is greater than 3.82 MPa, experimental stiffness is slightly lower than the numerical simulation value of this model. When normal pressure is 1.91 MPa, the numerical simulation value of the stiffness of this model is 2.43 MPa/μm and the experimental stiffness is 2.51 MPa/μm. The curve has slightly deviated, and the maximum relative error is 2.89%. The roughness is *Sa* 5.8 μm. When normal pressure reaches 2.29 MPa, experimental stiffness reaches 2.11 MPa/μm. The stiffness curve of this model intersects the experimental stiffness curve. When normal pressure is less than 2.29 MPa, experimental stiffness is slightly greater than the numerical simulation value of this model, and when normal pressure is greater than 2.29 MPa, experimental stiffness is slightly lower than the numerical simulation value of this model. When normal pressure is 5.09 MPa, the numerical simulation value of the stiffness of this model is 3.17 MPa/μm and the experimental stiffness is 3.08 MPa/μm. The curve has slightly deviated, and the relative error is up to 2.89%.

The above analysis shows that the stiffness curve value of this model is highly consistent with the experimental stiffness curve value, and the model has a certain application value. There is high agreement between the experimental stiffness values and the simulated model stiffness values under the four typical roughness conditions. Firstly, the new model in this paper assumes the asperities to be parabolic cylindrical asperities, which is more consistent with the actual contact situation. Secondly, according to the elastoplastic contact theory, the contact deformation of asperities is divided into the elastic deformation stage, elastoplastic deformation stage, and plastic deformation stage, which is closer to their real contact deformation state. Finally, the test device designed in this experiment is more reasonable and effective, and the test instrument is of high precision and has a higher reference value. The high coincidence between experimental stiffness values and simulated model stiffness values under four typical roughness conditions indicates that the model is not affected by various roughness conditions, which proves that the calculation method of the model is real and reliable.

The above experimental stiffness data are the average values of stiffness obtained through ten experiments. Figure 9 shows a bar chart of relative error of the experimental stiffness results of four roughness samples compared with the stiffness of the new model in ten experiments, with a pressure of 6.3 MPa. The relative error of ten experiments is less than 4.3%, which proves that the experimental results are reliable and stable with high repeatability.

To verify the superiority and application of the presented model, the presented model, the GW model, the KE model, the CEB model, and experimental results were compared and analyzed. The comparison results of contact stiffness under different roughness conditions are shown in Figure 10.

Figure 10 shows the comparison between the experimental curves of normal contact stiffness of surfaces with different roughness and the stiffness curves of this model, the GW model, the CEB model, and the KE model. Figure 10 shows the relative errors between the experimental values of normal contact stiffness of surfaces with different roughness and the stiffness values of the presented model, the GW model, the CEB model, and the KE model.

As can be seen from Figure 10a, when roughness is *Sa* 1.6 μm, with an increase in normal contact pressure, the curve of this model and the experimental curve have the same trend, and the simulation value of normal contact stiffness is close to the experimental value. However, the stiffness curves of the GW model, the CEB model, and the KE model gradually deviate from the experimental stiffness curve, and the difference becomes greater and greater. As can be seen from Figure 11a, when roughness is *Sa* 1.6μm, the relative error of the normal contact stiffness value of the new model is between 0.303% and 2.56%, with a small difference. The relative error of the GW model is the greatest, reaching 157.9%, and the relative error remains unchanged during the whole loading process. When the pressure is 0.64 MPa, the relative error of the CEB model is 134%, and gradually decreases to 67.5% with an increase in normal contact pressure. When the pressure is 0.64 MPa, the relative error of the KE model is 90.3%, and gradually decreases to 24.2% with an increase in normal contact pressure. The relative errors of CEB and KE models gradually decrease because the two models considered elastoplastic deformation. With an increase in pressure, the asperities began to experience elastoplastic deformation, which made the two models closer to the experimental stiffness value compared with the GW model.

Figure 10b also shows the same curve trend as Figure 10a. When roughness is *Sa* 3.2 μm, with an increase in normal contact pressure, the model curve and the experimental curve have the same trend, and the simulation value of the normal contact stiffness is close to the experimental value. However, the stiffness curves of the GW model, the CEB model, and the KE model gradually deviate from the experimental stiffness curve, and the difference becomes greater and greater. As can be seen from Figure 11b, when roughness is *Sa* 3.2 μm, the relative error of the normal contact stiffness value of the new model is between 0.46% and 2.92%, with a small difference. The relative error of the GW model is the greatest, reaching 152.4%, and the relative error varies between 123.3% and 152.4% during the whole loading process. With an increase in normal contact pressure, the relative error value of the CEB model increases from low to high, and then gradually decreases. When the pressure is 1.91 MPa, the relative error reaches the greatest value of 108.4%. With an increase in normal contact pressure, the relative error of the KE model increases from low to high, and then decreases gradually. When the pressure is 1.91 MPa, the relative error reaches the greatest value of 75.1%.

Figure 10c also shows the same curve trend as Figure 10b. When roughness is *Sa* 4.5 μm, with an increase in normal contact pressure, the stiffness curve of this model and the experimental stiffness curve have the same trend, and the simulation value of normal contact stiffness is close to the experimental value. However, the stiffness curves of the GW model, the CEB model, and the KE model gradually deviate from the experimental stiffness curve, and the difference becomes greater and greater. As can be seen from Figure 11c, when roughness is *Sa* 4.5 μm, the relative error of the normal contact stiffness value of the new model is between 0 and 2.89%, with a small difference. The relative error of the GW model is the greatest, reaching 158.07%, and the relative error varies between 127.86% and 158.07% during the whole loading process. With an increase in normal contact pressure, the relative error of the CEB model gradually decreases from 68.4% to 43.78%, and that of the KE model gradually decreases from 46.13% to 17.56%.

As shown in Figure 10d, when roughness is *Sa* 5.8 μm, with an increase in normal contact pressure, the curve of this model and the experimental curve have the same trend, and the simulation value of normal contact stiffness is close to the experimental value. However, the stiffness curves of the GW model, the CEB model, and the KE model gradually deviate from the experimental stiffness curve, and the difference becomes greater and greater. As can be seen from Figure 11d, when roughness is *Sa* 5.8 μm, the relative error of the normal contact stiffness value of the new model is between 1.68% and 2.89%, with a small difference. The relative error of the GW model is the greatest: the highest value is 201.57%, and it varies between 126.8% and 201.57% during the whole loading process. With an increase in normal contact pressure, the relative error of the CEB model varies between 98.47% and 110.26%, and that of the KE model varies between 56.9% and 73.18%.

Through the comparison of the above models, it is found that the stiffness curve of the GW model has a high degree of deviation. The maximum error occurs because the GW model simplifies the combination of the rough surfaces to contact between a spherical asperity and the rigid plane. this is not the case for the actual asperities between the two rough surfaces.the contact process only considers the elastic deformation of the asperity and does not consider their elastic–plastic deformation. There is a single model calculation process, and the model is relatively simple. As the pressure increases, the deviation of the simulation stiffness value from the experimental stiffness value becomes greater. The error caused by ignoring plastic deformation is prominent, and the stiffness value deviates sharply from the experimental stiffness value. The KE model is a frictionless contact elastoplastic model between a deformable sphere and a rigid plate. The model considers the evolution of elastoplastic contact with an increase in pressure and reveals three different stages from completely elastic to elastoplastic and then to a completely plastic contact interface. It covers a wide range of normal contact stiffness values from the initial yield to a full plastic state in the spherical contact region. The model fully considers the influencing factors of plastic deformation, so the numerical simulation stiffness curve is smooth for the whole loading fitting process, and it rises with the trend of the experimental stiffness curve. Compared with the GW model curve, it is closer to the experimental stiffness value, and the error is relatively small. However, the KE model assumes that the asperity shape is a spherical convex body, and the contact between the two rough surfaces is equivalent to the contact between the smooth rigid plane and the equivalent rough surface, which does not accord with the actual contact of the rough surface. As a result, there is a certain error in the comparison between the simulated normal-contact stiffness curve and the experimental stiffness curve. The CEB model is an elastoplastic model for analyzing the contact of rough surfaces. The CEB model not only considers the elastic–plastic deformation characteristics of asperities, but also adds the volume conservation factor of asperities into the plastic deformation process. During the whole loading experiment, compared with the GW model curve, the stiffness value curve of the numerical simulation was smooth and increased with the experimental stiffness value curve. However, the CEB model assumed that the asperities were spherical and in contact with the rigid surface. The model topography assumption was inconsistent with the actual situation and deviated from the real contact law, which interfered with the simulation data results. It caused error in the numerical simulation stiffness value and the experimental stiffness value, so the accuracy of the model is poor. The model proposed in this paper first fits the contour parameters and topography characteristics of a single asperity by collecting the surface data of the rough body. Then, the simulation surface of the rough body is reconstructed according to the fitting data, and the parabolic Y = CX^2^ +GX + H cylinder cross normal contact stiffness model is established. Then the pressure and normal displacement of the asperity in the elastic deformation stage, elastoplastic deformation stage, and complete plastic deformation stage were analyzed. Finally, the normal contact stiffness of the whole joint surface was obtained based on statistical theory. Therefore, the numerical simulation stiffness curve of this model is consistent with the experimental stiffness value. According to the above comparison results, the new model proposed in this paper is more accurate than the GW, the KE, and the CEB models.

## 6. Conclusions

A novel analytical model of normal contact stiffness for mechanical joint surfaces was proposed in this paper. The main conclusions can be summarized as follows:

A hypothetical surface with a higher degree of agreement with the actual surface is established. Based on the fitting of actual machined surface-topography data, a hypothetical surface with parabolic cylindrical asperity and Gaussian distribution is proposed. In this hypothetical surface, a parabolic column is used to simulate the shape of a single asperity surface, and the height distribution of the asperity is characterized by Gaussian distribution.A new normal contact stiffness model is established. Based on the hypothetical surface and combined with contact mechanics analysis of the rough surface, a general normal contact stiffness model for plane-cutting surfaces was established. This model recalculates the relationship between indentation depth and contact force in the elastic, elastoplastic, and completely plastic deformation regions of asperities, and finally obtains an analytical model of normal contact stiffness.The accuracy of this model is verified by experimental method. To verify the accuracy of the proposed method, a contact stiffness test bed was built, and the model in this paper was compared with the experimental results. The comparison results show that the model in this paper is consistent with the experimental results, which verifies the rationality and correctness of the proposed method. In addition, the proposed model is compared with the GW model, the CEB model, and the KE model in terms of the fitting data of normal contact stiffness. The results show that when roughness is *Sa* 1.6μm, the maximum relative errors are 2.56%, 157.9%, 134%, and 90.3%, respectively. When roughness is *Sa* 3.2 μm, the maximum relative errors are 2.92%, 152.4%, 108.4%, and 75.1%, respectively. When roughness is *Sa* 4.5 μm, the maximum relative errors are 2.89%, 158.07%, 68.4%, and 46.13%, respectively. When roughness is *Sa* 5.8 μm, the maximum relative errors are 2.89%, 201.57%, 110.26%, and 73.18%, respectively. The results show that the model in this paper is closer to the experimental results than other models, thus verifying the accuracy of the model in this paper. The normal contact stiffness of mechanical joint surfaces is of great significance to the research and analysis of mechanical contact characteristics. Attention should be paid to calculations for contact of engineering surfaces. This paper mainly considers the influence of normal contact stiffness on contact calculations. In fact, tangential contact stiffness, lubrication, vibration damping, dynamic characteristics, and other factors should also be considered in the study of the contact characteristics of mechanical joints. The influence of these factors will be examined in future studies.

## Figures and Tables

**Figure 1 materials-16-01883-f001:**
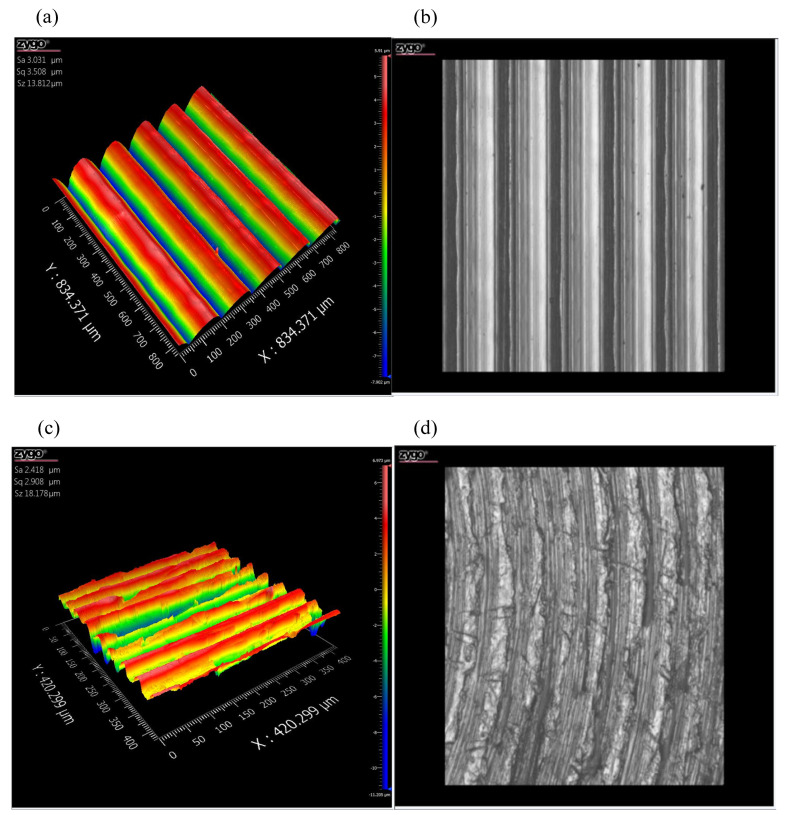
The surface topography under different machining methods: (**a**) boring; (**b**) real image of boring; (**c**) turning; (**d**) real image of turning; (**e**) smooth grinding; (**f**) real image of smooth grinding; (**g**) grinding; (**h**) real image of grinding.

**Figure 2 materials-16-01883-f002:**
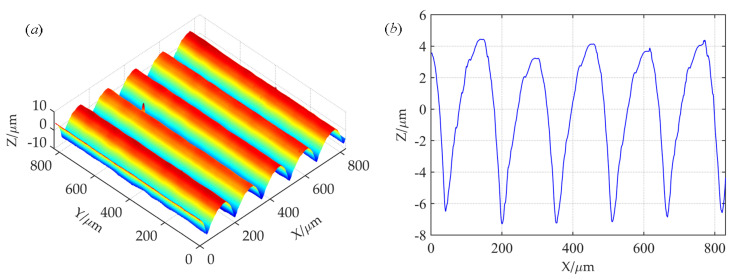
Reconstructed topographies in boring machining method: (**a**) 3D reconstructed topography; (**b**) 2D reconstructed topography.

**Figure 3 materials-16-01883-f003:**
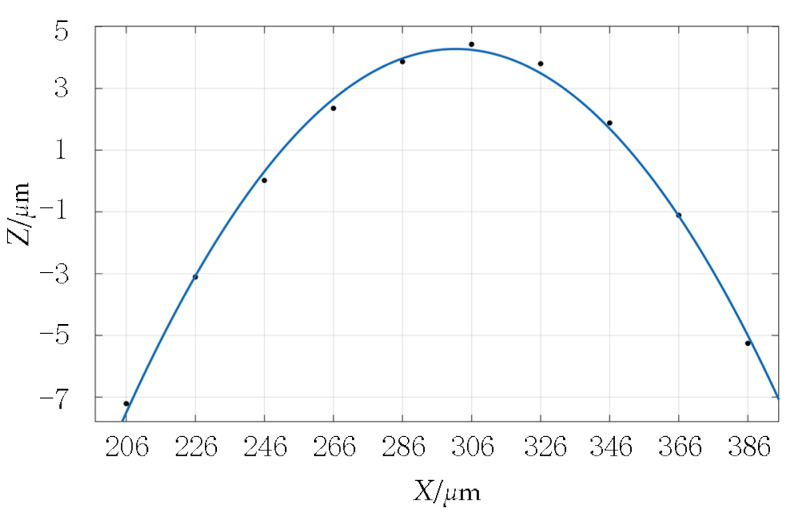
The fitting result of a single asperity.

**Figure 4 materials-16-01883-f004:**
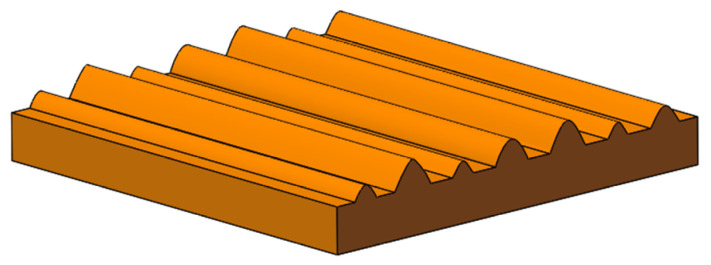
The hypothetical surface.

**Figure 5 materials-16-01883-f005:**
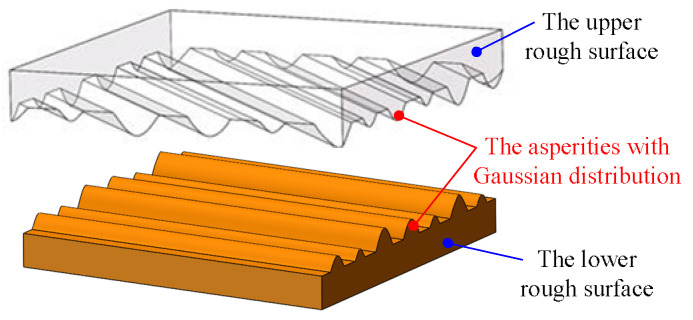
The schematic diagram of the contact model.

**Figure 6 materials-16-01883-f006:**
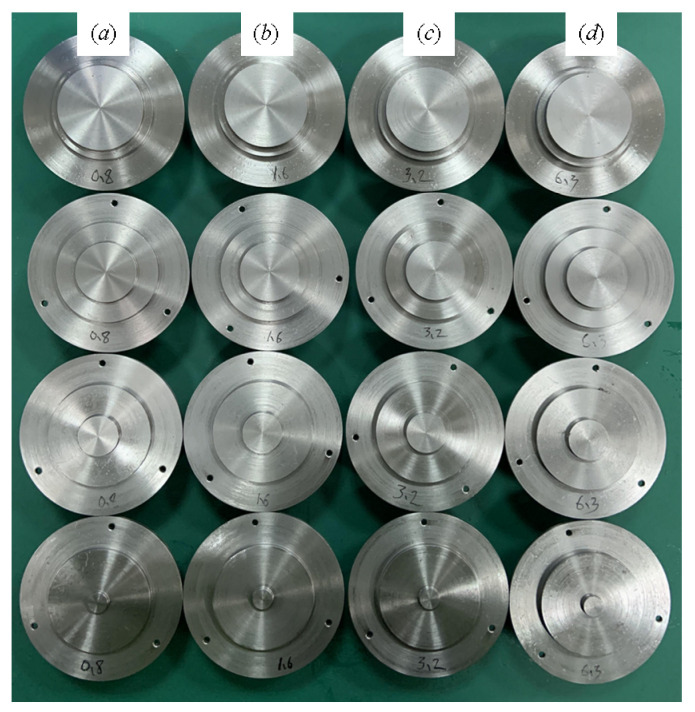
Surface to be tested: (**a**) *Sa* 1.6 μm; (**b**) *Sa* 3.2 μm; (**c**) *Sa* 4.5 μm; (**d**) *Sa* 5.8 μm.

**Figure 7 materials-16-01883-f007:**
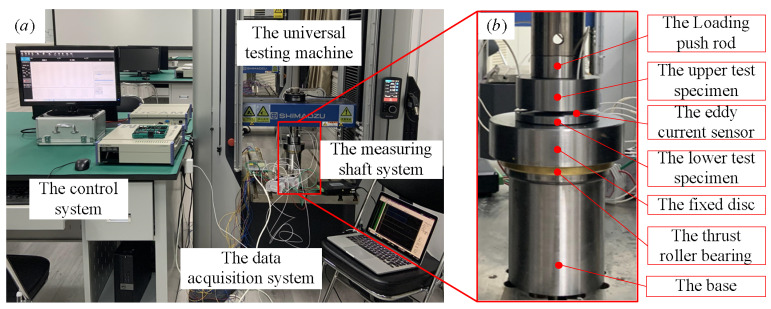
The experimental device: (**a**) the whole set of devices; (**b**) the measuring shaft system.

**Figure 8 materials-16-01883-f008:**
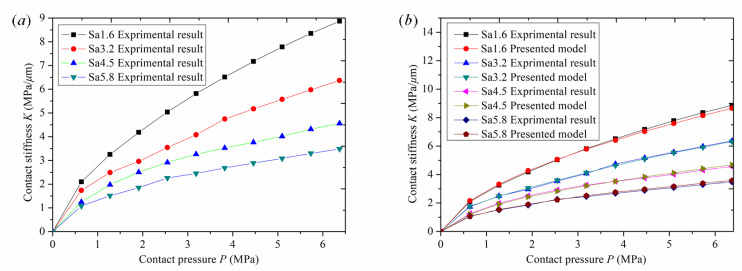
Normal contact stiffness: (**a**) the experimental results; (**b**) comparison between experimental results and the proposed model.

**Figure 9 materials-16-01883-f009:**
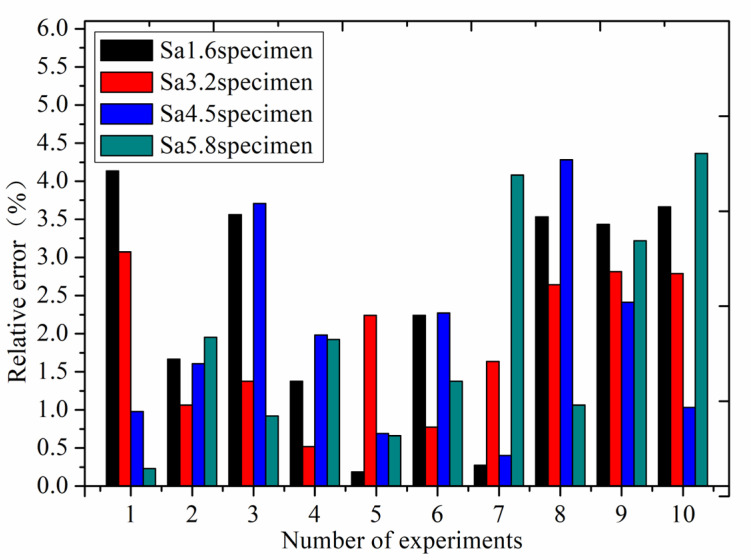
The relative error of ten experiments.

**Figure 10 materials-16-01883-f010:**
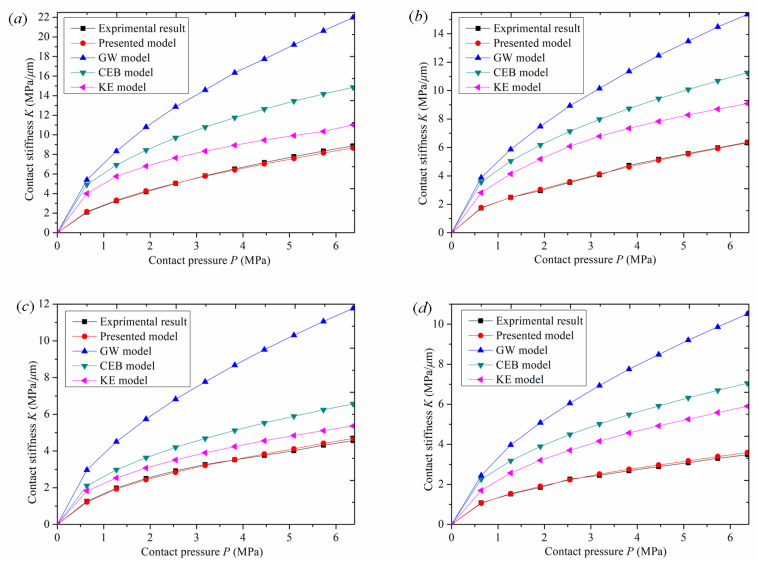
Comparison results of contact stiffness: (**a**) *Sa* 1.6 μm; (**b**) *Sa* 3.2 μm; (**c**) *Sa* 4.5 μm; (**d**) *Sa* 5.8 μm.

**Figure 11 materials-16-01883-f011:**
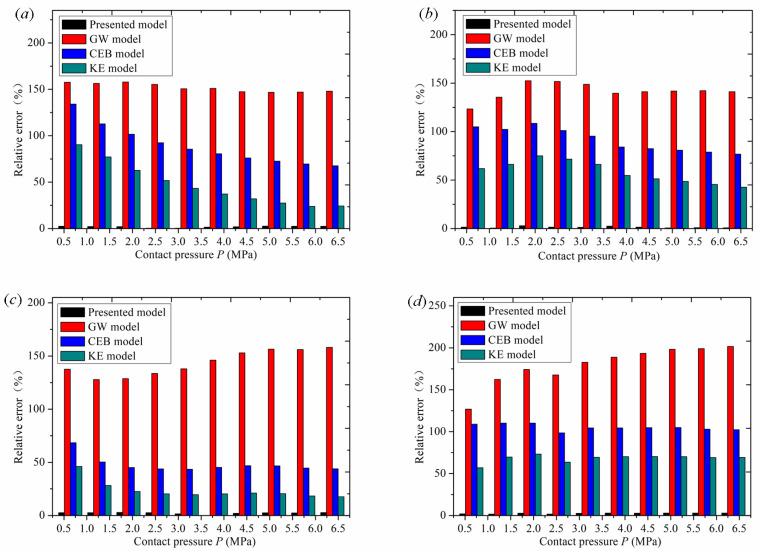
Relative error of contact stiffness: (**a**) *Sa* 1.6 μm; (**b**) *Sa* 3.2 μm; (**c**) *Sa* 4.5 μm; (**d**) *Sa* 5.8 μm.

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
