# Peer review of "An Analytical Model for the Normal Contact Stiffness of Mechanical Joint Surfaces Based on Parabolic Cylindrical Asperities"

_materials, 2023, doi:10.3390/ma16051883_

Round 1

Reviewer 1 Report

In this work, the authors have done nice work comparing the numerical simulation results with the experimental results using the GW model, CEB model, and KE model.  

It can be accepted after the major revision considering the following points. 

Abstract

1. write the full form of GW, CEB, and KE in the abstract. 

2. Add the comparative results between the numerical simulation results and the experimental results in percentage.

3. Also At present should be changed to "In the present study"

Introduction

1.  Add some relevant citation of reference in the first paragraph of the introduction

2.  Literature review part needs to be rewritten with brief information only. write the critical observations only. Add more literature.

3. state the novelty in the last paragraph of the Introduction.

Establishment of the hypothetical surface 1. Add the real images of turning, boring, and grinding with the sample image. 2. What is the difference between smooth grinding and grinding?   Contact stiffness analysis of a single asperity   1. Add the references for all equations mentioned in this section.   2. How you measured the intersection Angle φ of the axes of two asperities is φ = 45°?   The experimental part    1. Change this heading to "Experimental assessment" 2. Which operation was used for the experimentation? state the machining parameters.     Compare results and discussion 1. Change the heading to " Comparative results and discussion" 2.  Add the agreement and disagreement for the sentence "it can be seen that the stiffness of the joint surface with small roughness is always greater than that of the joint surface with large roughness as the force increases, and the normal contact stiffness also increases with the increase of the normal contact force."  3. Add the citation for this statement "The contact deformation of the joint surface is mainly elastoplastic, and the relationship between normal force and normal stiffness shows a nonlinear trend." 4.  Overall, I can state that the authors have not included the agreement and disagreement considering the literature in the discussion part. Therefore, I recommended adding this.   Conclusions The Conclusion should be specific. Not like a paragraph. Write specific points only. Also, add the future aspects of this study which will be useful for the reader.    All Figures are good.  

Author Response

We thank the reviewers for his careful read and thoughtful comments on the previous manuscript. We have carefully taken his comments into consideration in preparing our revision, which has resulted in a paper that is clearer, more compelling, and broader. The following attached pdf is the detailed modification.

Reviewer 2 Report

There are some English usage errors and too many paragraphs with only one sentence. The quality of some figures needs to be improved. Otherwise it is a good paper.

Author Response

(The authors gave the same response as above.)

Reviewer 3 Report

The paper describes an analytical solution for determining the contact stiffness of surfaces with parabolic asperites lying at a defined angle to each other. The results are compared with measurements and existing established models. The paper is written in a very understandable way. Nevertheless, I have the following comments that the authors should comment on.

1) Is Tabor's statement that the transition from elastic behaviour starts at 0.4H an estimate? I would imagine that such a statement depends on many kinematic and material properties.

2) As I understand the measurements, they were performed only once. I therefore find the significance of the experimental results to be limited.

3) The stated material parameters of the Young's modulus, the Poisson's ratio and the hardness are usually associated with uncertainties that may well be +-5%. This information is missing for me to be able to evaluate the agreement between measurement and model.

4) I do not think it is quite fair to compare a model for cylinders (the one presented by the authors) with a model for spheres when the surfaces are more similar to the first. Perhaps a comparison with other 2D models would be more helpful here. The authors do discuss this point in the Results, but I don't see the explanation for this in the Conclusions.

5) As I understand it, only experiments / models were used here in which the asperites have the same lateral distance from each other, i.e. the wavelength is constant, right? I could imagine that the model loses quality if more stochastic arrangements (with regard to the asperity distance) are also investigated. This point should at least be briefly discussed by the authors.

Author Response

(The authors gave the same response as above.)

Reviewer 4 Report

This paper presents a contact model based on parabolic cylindrical asperity and the cross cylinder Hertzian contact theory. Results of the normal contact stiffness for the mechanical joint surface are obtained and validated with the experimental test. There are some errors in the equations and information is missing. Please see my comments below and the attached PDF file.

·        What are the values of parameters C, G and H particularly for the curve-fit in Fig.3?

·        What is the height range in numerical models and in the experimental tests?

·        Eq.(7)->Eq.(8): as A+B = R, R should be the numerator in Eq.(8) rather than the denominator. So it is incorrect in Eq.(9) and (10). Please add a reference to Eq.(7).

·        How sensitive the cross-section profile of the asperity is? What if using a circle or elliptical shape to represent the cross-section of asperities and what could be the errors? Can you include any literature or add a section on analytical solutions?

·        Can the authors provide more results on the contact area and contact stress? A universal model should be able to present a full numerical solution to a contact problem apart from the stiffness.

·        Please provide the equations used to calculate normal contact stiffness in the GW, KE, CEB models, as this would be helpful in understanding the difference between these models.

·        A nomenclature is required to list variables and parameters used in equations. Some variables are not explained in the context of equations. Please see more details in the attached file.

·        Page 14, can you specify the deformation stages at certain loads, i.e., at which load does the elastoplastic and purely plastic deformation start to occur?

·        The discussion section in page 16-17 is a bit too wordy and repetitive which is not necessary, as Fig.9 has clearly stated. It could be better to highlight the main features and differences between the GW, KE, CEB models in a more structured way, such as sub-sections or a table, to list what is simplified in geometry, contact or elastic/plastic deformations.

·        While this model doesn’t consider the bulk deformation of the surface, the authors could provide an approximation of the bulk deformation which should be much smaller compared to the deformation of asperities to validate the model.

Author Response

(The authors gave the same response as above.)

Reviewer 5 Report

Nomenclature must be included in the paper. Abbreviations cannot be used in abstract, only full names - please correct line 21 of the article. Each symbol, abbreviation should be included in the nomenclature. Nomenclature can be both at the beginning of the paper or at the end of the manuscript.

The introduction is good, a proper review of the literature.

In relation to a scientific paper - a scientific article, please do not use the word "work" - preferred words "paper, manuscript, scientific article".

Research material - laboratory specimen - the test specimen, should be referred to as "specimen" - I do not recommend the word "sample". Please correct it.

Please prepare lines 225 - 232 in the form of a numbered list.

It seems to me that the hypothetical equation given in line 226 should also be numbered - this requires renumbering subsequent equations in the paper. Unless the authors articulate the equation differently.

Lines 253 - 265 of the thesis should also be reworded - it seems to me that the thesis needs to be changed. The authors need to redo their paper a bit.

Please check some mathematical entries in the text of the manuscript - wrong position of mathematical formulas in relation to the text.

Line 340 - please specify the chemical composition of steel 45. What are the equivalents of this steel according to ASTM, BS, ISO standards? Please mention it at manuscript. Please add table with chemical composition.

What is the unit of hardness H=1970MPa? Is it “MPa”? I would give the hardness in the Brinell, Rockwell or Vickers scale.

Please add parameterized technical drawings of specimens with dimensions to the paper. It is necessary.

In the research part, it is necessary to indicate what signals the authors recorded during the tests, what were the physical quantities, what were the units. Please add to the paper sample figures that illustrate the signals recorded during laboratory tests.

These should be comparative figures so that readers know the reproducibility of the research.

Please indicate how many specimens for each roughness and diameter were tested by the authors - please specify the repeatability of measurements, convergence of results - this should be included in the manuscript. This is not due to paper at the moment. Please include this in your manuscript.

Figure 6 shows that the authors had 4 specimens for each roughness group. For each of these groups, there should be comparative charts in the paper - this is not in the paper. Please correct the paper and resubmit it for review.

I suggest a major revision.

Author Response

(The authors gave the same response as above.)

Round 2

Reviewer 1 Report

The authors have done all required corrections in the revised manuscript. Now it can be accepted. 

Author Response

We thank the reviewers for his careful read and thoughtful comments on the previous manuscript. We have carefully taken his comments into consideration in preparing our revision, which has resulted in a paper that is clearer, more compelling, and broader.

Reviewer 3 Report

Dear authors,
Thank you very much for the responses to my comments. I must admit, however, that further questions now arise for me. If you have already carried out more than 10 measurements, you should definitely enter an error bar with the measurement results. I think that is self-evident. Your results suggest that the model matches the measurements almost perfectly. The quality of the results must in any case be evaluated in relation to the model uncertainties (e.g. Tabor's 0.4H assumption) and material uncertainties. This aspect is still completely missing for me.
In my opinion, your contact problem has also similarities with a 2D problem  (although due to the relative twisting of the surfaces the problem is of course 3-dimensional). For this reason, I would have thought it very appropriate to also add a 2D model as another reference.
Your statement that the slight fluctuations of the peaks also affect the wavelength is not convincing to me (as the amplitude only slightly changes the wavelength. Since the counterbody has similar wavelengths, I don't think the contact problem is really stochastic in this form. In fact, you do have a Gaussian distribution of the height coordinates, but that is only one of many criteria for characterising a stochastic surface.
In summary, I would say that you are not critical enough of your own results. You should, in my opinion, significantly expand the associated discussion of the results.

Author Response

(The authors gave the same response as above.)

Reviewer 5 Report

The authors included all my suggestions in the revised version of the paper. I recommend the manuscript for publication.

Author Response

(The authors gave the same response as above.)
